# The Anti-Biofilm Potential of Linalool, a Major Compound from *Hedychium larsenii*, against *Streptococcus pyogenes* and Its Toxicity Assessment in *Danio rerio*

**DOI:** 10.3390/antibiotics12030545

**Published:** 2023-03-09

**Authors:** Sarath Praseetha, Swapna Thacheril Sukumaran, Mathew Dan, Akshaya Rani Augustus, Shunmugiah Karutha Pandian, Shiburaj Sugathan

**Affiliations:** 1Department of Biotechnology, Kariavattom Campus, University of Kerala, Thiruvananthapuram Pin-695 581, Kerala, India; 2Department of Botany, Kariavattom Campus, University of Kerala, Thiruvananthapuram Pin-695 581, Kerala, India; 3Plant Genetic Resource Division, Jawaharlal Nehru Tropical Botanic Garden & Research Institute, Palode, Thiruvananthapuram Pin-695 562, Kerala, India; 4Department of Biotechnology, Alagappa University, Karaikudi Pin-630 003, Tamil Nadu, India

**Keywords:** *Streptococcus pyogenes*, biofilm, *Hedychium larsenii*, *Danio rerio*, AlamarBlue^TM^ assay, linalool

## Abstract

The anti-biofilm and anti-virulence potential of the essential oil (E.O.) extracted from *Hedychium larsenii* M. Dan & Sathish was determined against *Streptococcus pyogenes*. A crystal violet assay was employed to quantify the biofilm. Linalool, a monoterpene alcohol from the E.O., showed concentration-dependent biofilm inhibition, with a maximum of 91% at a concentration of 0.004% (*v*/*v*). The AlamarBlue^TM^ assay also confirmed Linalool’s non-bactericidal anti-biofilm efficacy (0.004%). Linalool treatment impeded micro-colony formation, mature biofilm architecture, surface coverage, and biofilm thickness and impaired cell surface hydrophobicity and EPS production. Cysteine protease synthesis was quantified using the Azocasein assay, and Linalool treatment augmented its production. This suggests that Linalool destabilizes the biofilm matrix. It altered the expression of core regulons *covRS*, *mga*, *srv*, and *ropB*, and genes associated with virulence and biofilm formation, such as *speB*, *dltA*, *slo, hasA*, and *ciaH*, as revealed by qPCR analysis. Cytotoxicity analysis using human kidney cells (HEK) and the histopathological analysis in *Danio rerio* proved Linalool to be a druggable molecule against the biofilms formed by *S. pyogenes.* This is the first report on Linalool’s anti-biofilm and anti-virulence potential against *S. pyogenes*.

## 1. Introduction

*Streptococcus pyogenes*, or β-hemolytic Group A *Streptococcus* (GAS), is an exclusively human pathogen with greater adaptabilities. It causes mild to severe life-threatening infections. Life-threatening “post-infectious non-pyogenic” diseases, such as rheumatic fever and acute glomerulonephritis, result from constant GAS infections [1]. *S. pyogenes* is equipped with various virulence factors, such as M proteins, hyaluronic acid capsule, cysteine proteases, pyrogenic exotoxins, and biofilm formation [2]. The emergence of antibiotic resistance caused by *S. pyogenes* strains is becoming a global health problem. The capacity of *S. pyogenes* to build a biofilm is one of the leading causes of antibiotic resistance.

Thanks to the rapid development of materials and microelectronics in implantable medical devices, many different diagnostic and therapeutic alternatives are available to human society. Although life has improved and changed in innumerable ways because of these breakthroughs, introducing implanted devices into the human body fosters microbial colonization and illnesses [3,4]. These gadgets attract multidrug-resistant bacteria that quickly cover them in biofilm mesh [5]. The term “biofilm” refers to the microbial consortium adhering to organic or inorganic surfaces using polysaccharides, proteins, and nucleic acids secreted by them. Biofilm development on indwelling medical equipment, such as catheters, heart valves, and mechanical ventilators, is responsible for the pathophysiology of nosocomial infections [6]. In addition to the healthcare industry, biofilm development affects many other sectors, such as the nautical, aquaculture, food, and dairy industries [7]. The biofilm framework’s structural integrity is provided by exo-polysaccharide (EPS), a biofilm matrix component that is a barrier to drugs and other protective substances [8]. The breakdown and elimination of EPS are problematic in the medical profession since microbes rapidly create strong and dense biofilms on medical devices [9]. Surfactants and strong chemical disinfectants do not affect biofilm removal; they might damage delicate medical equipment, such as endoscopes. This issue can be resolved by destabilizing the biofilm architecture using effective dispersal agents [10]. Anti-biofilm substances prevent bacteria from adhering to surfaces without affecting their growth or metabolism and slow the spread of antibiotic resistance [11]. *In vitro* studies have shown the potential of natural compounds as effective agents to combat biofilm-forming pathogens. 

To treat various microbial pathogens that cause diverse human diseases, essential oils (E.O.s) found in many herbal plants have also been employed as natural treatments. Most phytoalexins are substances with solid bacteriostatic or bactericidal action in E.O.s. Complex combinations of bioactive substances, such as monoterpenes, sesquiterpenes, or diterpenes, are found in E.O.s [12]. Although studies on the antibacterial activities exhibited by various E.O.s against different microbial pathogens have been documented, the growth inhibition and anti-biofilm actions of these E.O.s against *S. pyogenes* have not been thoroughly examined. 

One of the most attractive genera in the Zingiberaceae family is *Hedychium*, which is popularly known as butterfly lilies. Some *Hedychium* species are used in traditional medicine to treat gastrointestinal illnesses, bronchitis, asthma, blood purification, and antiemetics [13]. The present study deals with the anti-biofilm activity of an active lead compound, Linalool (Figure 1), from *Hedychium larsenii* against *S. pyogenes.* Linalool (3,7-dimethyl-1,6-octadien-3-ol) is an acyclic monoterpene alcohol found in many plant essential oils. It is generally recognized as safe (GRAS) and used as a food additive. Linalool has many biological properties, including anticonvulsant, analgesic, anti-inflammatory, sedative, anesthetic, anxiolytic, antioxidant, and antimicrobial properties [14]. Compared to the above-stated biological properties, studies on the anti-biofilm activity of Linalool and its mechanism of action are limited. The present study aimed to evaluate anti-biofilm activity and unravel the potential mechanism of action through a transcriptional approach. The *in vitro* findings were further corroborated by *in vivo* studies using an experimental model organism, *Danio rerio,* showing the non-toxic nature of the compound. 

## 2. Results

### 2.1. GC-MS Analysis

GC-MS analysis of *H. larsenii* showed the presence of 22 compounds (Table 1). A few of them were α-Pinene (1.02%), β-Pinene (3.72%), Cymene (11.77%), Linalool (52.11%), Linalool Oxide B (2.87%), α-Terpineol (2.95%), and Carvacrol (0.26%). Linalool was found to be the principal constituent of this essential oil. 

We primarily relied on Robert Adams’ mass spectral libraries to calculate relative retention indices (RRI) [15]. The RRI obtained was compared with the Robert Adams index, and mass spectra (M.S.) of the compounds obtained through Wiley 8 and the NIST 11 library, which were compared in conjunction with Adams’ library.

### 2.2. Minimum Inhibitory Concentration (MIC) of Linalool against S. pyogenes

The broth microdilution method and resazurin assay were used to determine the MIC of Linalool against *S. pyogenes.* At 0.01% (*v*/*v*), Linalool inhibited all visible growth of *S. pyogenes.* As an ideal anti-biofilm agent should not impede the development of the bacterium under study, concentrations below 0.01% were selected to evaluate Linalool’s anti-biofilm potential.

### 2.3. Non-Antibacterial Anti-Biofilm Activity of Linalool against S. pyogenes

The anti-biofilm potential of Linalool against *S. pyogenes* was investigated by 24-well plates following a crystal violet assay. The results showed a concentration-dependent anti-biofilm activity of Linalool with a maximum biofilm inhibition of 91% at a concentration of 0.004% (*v*/*v*). Concentrations above 0.004% (*v*/*v*) did not significantly increase biofilm inhibition. Hence, this was fixed as the MBIC (Minimum Biofilm Inhibitory Concentration) (Figure 2). Thus, all further assays were performed at this MBIC concentration. To prove that the observed biofilm suppression was not the result of growth inhibition, a growth examination employing spectrophotometry revealed no appreciable differences between the control and Linalool-treated samples (Figure 2).

### 2.4. Effect of Linalool on Growth and Cell Viability

The CFU and cell viability assays were performed to assess the effect of Linalool on the growth and metabolism of *S. pyogenes*. The AlamarBlue^TM^ cell proliferation assay was carried out to determine the impact of Linalool on cell viability and cytotoxicity. No significant change in the percentage reduction of AlamarBlue^TM^ between the treated and control cells was seen. Compared to the control samples, the decrease in Linalool-treated samples was 97.38% (Figure 3a). In addition, no significant reduction was observed in CFU values of Linalool-treated cells (Figure 3b). Thus, it is inferred that cells are metabolically active when treated with Linalool at its MBIC of 0.004%, proving that the compound under investigation is non-bactericidal and non-bacteriostatic. 

### 2.5. Microscopic Observation Confirms the Anti-Biofilm Potential of Linalool

When treated samples were compared to their untreated control counterparts, a light microscopic examination of the biofilms generated in the presence and absence of Linalool (at 0.004%) demonstrated a decrease in the surface area covered by biofilms (Figure 4A). The light micrographs showed that the control samples were more decadent with thick biofilms. A CLSM investigation was conducted to support the anti-biofilm potential of Linalool. The mature biofilm architecture and the micro-colonies were efficiently reduced in the tested strain compared to control slides with dense micro-colonies (Figure 4B). In the treated sample, *S. pyogenes* cells were seen as a chain-like structure as opposed to the control surface, which was coated in biofilm. There was a reduction in surface coverage in Linalool-treated glass slides. SEM analysis was performed to confirm whether Linalool treatment caused cell surface modification in *S. pyogenes.* The control sample showed an aggregated form of bacterial chains, with sticky EPS encircling the cells. Linalool-treated samples had a clear cell surface and dispersed chains in contrast to the control, indicating decreased EPS production (Figure 4C). The findings were further supported by EPS quantification using the phenol-sulfuric acid method, which revealed an 85% reduction in EPS in the treated sample compared to the control (Figure 5A).

### 2.6. Linalool Mitigates EPS and Cell Surface Hydrophobicity

EPS aids in several processes, such as initial attachment, biofilm architecture maturing, and enhancing biofilms’ mechanical durability. EPS quantification was done using the phenol–sulfuric acid method. Treatment with Linalool (0.004%) reduced EPS production by 75% (Figure 5A). A crucial characteristic that enables *S. pyogenes* to attach to polar and nonpolar surfaces and to form biofilms is cell surface hydrophobicity (CSH). Control cells had a cell surface hydrophobicity of 92%, whereas Linalool-treated cells had a reduced cell surface hydrophobicity of 48% (Figure 5B). 

### 2.7. Linalool Enhances Extracellular Cysteine Protease Production

An azocasein assay was used to quantify the extracellular cysteine protease production by *S. pyogenes*. Cysteine protease is a major secreted protease of *S. pyogenes,* and its synthesis directly correlates with the amount of colored azodye released by the enzyme. At 0.004% (MBIC), Linalool enhanced cysteine protease production (Figure 5C).

### 2.8. Dynamics in the Expression of Candidate Virulence Genes by Linalool

Real-time PCR analysis was carried out to determine the impact of Linalool on key genes associated with virulence factors and biofilm formation. qPCR results showed significant reduction in expression of *mga* (3.6-fold), *col370* (3.8-fold), *dltD* (3.8-fold), *hts* (4.9-fold), and *hasA* (3.5-fold). Moderate downregulation was observed in the expression levels of *covR* (2.4-fold), *covS* (1.9-fold), *slo* (1.2-fold), *srv* (2.3-fold), and *ciaH* (1.2-fold). On the other hand, the expression levels of *speB, ropB*, and *luxS* were upregulated by 2.4-, 2.4-, and 1.4-fold, respectively (Figure 6).

### 2.9. Cytotoxicity Analysis in HEK Cells 

The results of anti-biofilm suppression of Linalool necessitated the need for their cytotoxicity testing. Human Embryonic Kidney (HEK) cell lines were used for cytotoxicity testing. Cytotoxicity analysis was based on measuring MTT-mediated formazan production. Treatment with Linalool showed no reduction in cell viability. Linalool maintained the viability of the cells at 98% at the MBIC (Figure 7A). From the microscopic analysis, HEK cells were healthy and normal in Linalool-treated samples at MBIC (0.004%) (Figure 7B).

### 2.10. In Vivo Efficacy of Linalool in Danio Rerio

Zebrafish (*Danio rerio*) is a tropical fish species. It is regarded as one of the most versatile model organisms for *in vivo* studies due to its many experimental benefits, including its low cost, small size, and ease of laboratory maintenance [16]. To identify the non-lethal concentrations, toxicity testing of Linalool was performed in Zebrafish (Figure 8A) under OECD Guideline 203. The survival rate of Zebrafish in the range of 0 to 96 h at the concentration of MBIC×, MBIC × 2, MBIC, and MBIC/2 was analyzed (Figure 8B). This study proved that Linalool was non-toxic to Zebrafish up to the MBIC × 2 concentration. At the MBIC × 4 concentration, only 48% of fish survived, showing that its toxicity surged at higher concentrations. As Linalool was non-toxic up to MBIC × 2, three concentrations were taken, MBIC/2 (0.002%), MBIC (0.004%), and MBIC × 2 (0.008%) for histopathological studies. To analyze the effect of Linalool on the morphology of the vital organs, fish were euthanized after 96 h, and the histopathology observation of vital organs, such as gills (Figure 8C), kidney (Figure 8D), and liver (Figure 8E), were examined, and all showed normal anatomy among the blank control, solvent control, and at concentrations of MBIC × 2, MBIC, and MBIC/2 of Linalool.

## 3. Discussion

Diseases caused by *S. pyogenes* frequently do not respond to antibiotic therapy because of their ability to produce a biofilm *in vivo* [17]. Infections caused by medical equipment are complicated to treat because of host phagocytic evasion, biofilm formation, and drug resistance. Agents that can prevent *S. pyogenes* biofilm formation are needed. In the present study, the essential oil obtained from the fresh rhizomes of *Hedychium larsenii* was screened for its anti-biofilm capacity. GC-MS analysis showed Linalool to be the primary component present in the essential oil. Linalool was screened for its ability to inhibit biofilm formation in *S. pyogenes*. Linalool is one of the main ingredients of fragrances, flavoring, and cosmetic products. The antimicrobial and insect-repelling properties of Linalool have been demonstrated widely [18]. Linalool has been used extensively in industries thanks to its superb antibacterial properties. It exhibited remarkable antibacterial activity against *Shewanella putrefaciens* [19]. Its antimicrobial and anti-biofilm potential has been demonstrated against *Listeria monocytogenes* [14]. However, its anti-biofilm activity has never been explored against *S. pyogenes.* This study assessed the anti-biofilm potential of Linalool against *S. pyogenes* SF370.

Linalool exhibited a concentration-dependent biofilm inhibition of 91% at a 0.004% (*v*/*v*) concentration. In comparison, limonene—a cyclic monoterpene found in citrus fruits—demonstrates 80–90% biofilm suppression against several M serotypes at a significantly greater dosage of 400 μg/mL [20]. Eucalyptol exhibited 89% biofilm inhibition in *S. pyogenes* at a much higher concentration of 0.3% [21]. A flavonol called morin hydrate reduced biofilm development by 50% at 220 µM compared to the control [22]. The essential oil showed varied biofilm inhibitory activity from *Patchouli cablin* and *P. heyneanus* against various *S. pyogenes* strains. At the lowest concentration of 5%, 35–40% biofilm inhibition was reported, while at the highest concentration of 15%, 60–70% biofilm inhibition was exhibited by both oils [23]. Essential oils from *Origanum vulgarem* L. and *Salvia officinalis* L. inhibited biofilm formation at 0.5% [22]. Biofilm formation by *Candida albicans* and *C. dubliniensis* was reported to be inhibited by essential oil from *Mentha piperita* at concentrations as high as 2000 μg/mL [24]. Compared to the above anti-biofilm activities of other essential oils, Linalool exhibited better anti-biofilm inhibition at a lower concentration of 0.004%. 

Linalool exhibited no detrimental effect on the growth of *S. pyogenes* at its MBIC, as evidenced by CFU analysis and O.D. at 600 nm. The cell proliferation assay further proved that Linalool did not affect the growth and metabolism of *S. pyogenes.* For more than five decades, the AlamarBlue^TM^ assay has been used to assess the metabolic viability of cells as a gold standard technique. When exposed to living cells, the non-toxic, blue dye resazurin transforms into pink fluorescent dye resorufin through the action of oxidoreductases [25]. Thus, the viability assays proved the non-antibacterial nature of Linalool. The outcomes supported the notion that an ideal anti-biofilm agent would not be expected to provide any selection pressure on bacteria to prevent resistance emergence [26]. This takes Linalool one step closer to an ideal anti-biofilm agent. 

To study the effect of Linalool treatment on biofilm architecture created by *S. pyogenes*, microscopic studies of biofilms developed in the presence and absence of Linalool were carried out. Compared to dense biofilm with micro-colonies on the control surface, the treated surfaces, as shown by Light and CLSM microscopic images, showed a reduction in the micro-colonies. In the treated sample, *S. pyogenes* cells were seen as a chain-like structure as opposed to the control surface, which was covered with biofilm. Cell surface hydrophobicity is a crucial characteristic that enables *S. pyogenes* to cling to polar and nonpolar surfaces and enhances biofilm formation [25]. From visual inspection, it was clear that Linalool treatment changed the hydrophobicity of the cell surface. In Linalool’s presence, toluene treatment decreased cell partitioning toward the hydrophobic toluene phase. Thus, *S. pyogenes’* ability to adhere was compromised by a decrease in hydrophobicity after Linalool treatment. SEM analysis was done to verify that Linalool treatment resulted in cell surface change in *S. pyogenes*. In contrast to the control, in which the cells were coated with a polysaccharide matrix, Linalool-treated cells had a clean cell surface.

The presence of several virulence factors by *S. pyogenes* enables it to avoid the immune response, invade, and survive *in vivo*, and is always associated with its pathogenesis. Therefore, if these virulence components are attenuated, *S. pyogenes’* ability to infect will undoubtedly decrease. Microorganisms are known to create a variety of virulence factors, such as protease, which clears host extracellular matrix proteins; hemolysin, which cleaves human erythrocytes; and other immunological modulators for the progression of infection [25]. EPS serves many functions, such as helping in attachment; it acts as a cementing material that gives mechanical durability and strength to biofilms. EPS excludes the entry of antibiotics and thus results in antibiotic treatment failures and contributes to antibiotic resistance [27]. Images from SEM analysis showed that biofilm biomass and EPS appeared to be declining. The phenol-sulfuric acid method was used to quantify EPS, which showed decreased EPS production in the Linalool-treated sample. Reduced EPS following Linalool treatment ensures that *S. pyogenes* becomes more susceptible to antibiotics and is thus cleared by the host’s immune response. 

Cell surface hydrophobicity (CSH) makes it easier for bacteria to adhere to solid surfaces, which is a crucial stage in creating biofilms [28]. The MATH assay revealed that Linalool treatment significantly reduced the cell surface hydrophobicity of *S. pyogenes.* A component present in the cell walls of all Gram-positive bacteria is LTA, which is responsible for the cell surface hydrophobicity of the cell. *E. feacalis* and *S. aureus* have been shown to have impaired biofilm formation due to mutations in the *dltA* gene [29]. Linalool treatment downregulated the expression of the *dltA* gene, thus decreasing cell adherence. 

Streptococcal pyrogenic exotoxin B (SpeB) is a significant extracellular protease produced by *S. pyogenes*. It helps the bacteria by degrading vitronectin, matrix proteins, fibronectin, human interleukin-1b, and other immune cells, thus establishing infections and causing tissue destruction [30,31]. SpeB controls the dispersion and production of biofilms by destroying the peptides linked to the biofilm matrix [32]. Thus, it can be concluded that the enhanced protease synthesis observed after treatment with Linalool would prevent the formation of biofilms by proteolysis. Real-time PCR results also showed upregulation of *speB.* The expression of *speB* is regulated by the independent standalone regulators Srv and RopB. *speB* is positively regulated by RopB, but *speB* is negatively regulated by Srv [33,34]. Linalool increased the expression of *ropB* while downregulating *srv.* The combined effects of decreased CSH, elevated protease, and RT-PCR analysis show that numerous gene regulation pathways cause decreased biofilm formation after Linalool treatment. Another crucial factor for *S. pyogenes* survival is HtsA, a component of the streptococcal iron acquisition (SiaABC) transporter, which is involved in heme acquisition [35]. RT-PCR analysis showed that Linalool treatment significantly downregulated *htsA*. The *htsA* mutants showed defective iron uptake, affecting its virulence [36]. 

The current study’s repressor and sensor kinase genes of the covRS TCS pathway, covR, and covS, were downregulated. Nearly 15% of the genes in *S. pyogenes*, including *luxS* (involved in the quorum sensing system) *hasA* (hyaluronic acid synthesis) and *mga* (standalone regulator), are known to be regulated by the well-studied TCS covRS system [37]. Linalool treatment downregulated both *covR* and *covS*. *luxS* is involved in the quorum sensing process and in synthesizing AI-2, a quorum sensing molecule. Linalool treatment upregulated the *luxS* gene. The inactivation of *luxS* causes a reduction in *speB* production [38]. The upregulation of *luxS* thus confirmed enhanced *speB* production in our studies. Many genes were downregulated, such as *hasA*, *mga*, *slo*, *col*, and *srv*. These genes are involved in helping the pathogen adhere to host tissue (*col370*), elude the immune cells (*hasA*, *mga*, *slo*), and thereby establish chronic infections. 

The *in vivo* efficacy of Linalool was tested using *Danio rerio.* Zebrafish models are gaining importance for various reasons, including their preserved vertebrate biology, high fecundity, short generation time (2–3 months), simplicity of care, which allows for repeatability of experiments, and rapid embryonic development. Of its genes, 70% are orthologous to human genes [39]. Cytotoxicity was evaluated using the Zebrafish survival assay at 4×, 2×, 1×, and 0.5× MBIC of Linalool. Linalool at amounts of 0.008% (MBIC × 2), 0.004% (MBIC), and 0.002% (MBIC/2) showed no harmful effects on Zebrafish survival, and the survival rates of Linalool-dosed Zebrafish were comparable to those of the control fish. However, at 0.016% (MBIC × 4) of Linalool, the survival percentage of fish was reduced to 66% on the 2nd day and further reduced to 50% on the 3rd day. This proved that Linalool is non-toxic to Zebrafish up to 0.008% (MBIC × 2), and the toxicity increased at 0.016% (MBIC × 4). In a similar study, Myristic acid did not exert any toxicity on the survival of Zebrafish up to 125 μg/mL. While at 250 μg/mL, the survival rate was reduced to 40% on 3rd day, and all fish died on the 4th day of the dosage. Palmitic acid did not exert any toxicity in Zebrafish up to 400 μg/mL, and the survival rate was 100% [40].

A histopathological investigation was conducted to investigate the impact of Linalool on important organs, such as the gills, kidneys, and liver. All control groups (blank and solvent control) exhibited normal morphology. The gills are responsible for gas exchange in the water and excreting waste [41]. Toxin exposure causes common tissue changes in this organ, which are especially vulnerable to them. Upon exposure to Linalool at MBIC/2 (0.002%), MBIC (0.004%), and MBIC × 2 (0.008%), no apparent changes were seen in the gill structure of Zebrafish. As no change in the structure of the gills was noted in the treated samples, it proves that the compound is not toxic up to MBIC × 2 (0.008%). Zebrafish kidneys filter waste products, maintain osmotic balance, and eliminate extra water entering the mouth [39]. No noticeable change was seen in the morphology of the kidneys of the treated samples; Linalool proved safe. The hepatic cords are made up of hepatocytes, which are the main cells of the Zebrafish liver. Similar to higher animals, these cells aid in the digestion of proteins, lipids, carbohydrates, and vitamins. They have also been involved in the detoxification of xenobiotics. Identical to the gills and kidneys, Zebrafish treated with Linalool up to MBIC × 2 showed no substantial histopathological changes in the liver. In a similar study, treatment with 0.5 mg/l of maduramicin was found to cause significant changes in the gills, liver, and intestine structures, showing that it is toxic [42].

Linalool (C_10_H_18_O), also called 3,7-dimethyl-1,6-octadien-3-ol, is an unsaturated monoterpene alcohol with a distinct aroma, described as ‘slightly lemony.’ To highlight the present study’s findings, the plant terpene Linalool exhibited anti-biofilm properties and impaired the surface-associated virulence factors of *S. pyogenes*. Transcriptional studies revealed that Linalool targets *dltA*, *mga*, *hasA*, and *hts,* which affected *speB* and *ropB.* Linalool also impaired the virulence factors associated with invasiveness. Additionally, the non-toxic nature of Linalool in HEK cells and the *in vivo* histopathological analysis in *Danio rerio* mark Linalool as a promising therapeutic agent in treating biofilm-associated *S. pyogenes.* The current study is the first of its kind to demonstrate the *in vitro* and *in vivo* anti-biofilm potential of Linalool against *S. pyogenes*.

## 4. Material and Methods

### 4.1. Bacterial Strain and Growth Condition

*Streptococcus pyogenes* SF370 (ATCC700294D-5) was procured from the American Type Culture Collection (ATCC), USA. *S. pyogenes* in Todd-Hewitt broth (Hi-Media, Mumbai, India) supplemented with 0.5% yeast extract and 1% glucose (THYG broth) was used for the routine culture of *S. pyogenes*. Tryptone soy agar (Hi-Media, Mumbai, India) plates were used to store the organism in the refrigerator. Overnight culture of *S. pyogenes* in THYG broth (~2.3 × 10^5^ CFU/mL) was considered the standard cell suspension for biofilm studies [28].

### 4.2. Extraction of Essential Oil Using Hydro Distillation

Fresh rhizomes of the plant *Hedychium larsenii* were collected, washed, and cut into small pieces and subjected to hydrodistillation by a Clevenger-type apparatus for 4 h to obtain the essential oil. Accumulated essential oils were dried with anhydrous sodium sulfate and stored in an air-tight amber vial at 4 °C [43].

### 4.3. Phytochemical Analysis: GC-MS

The essential oil was subjected to gas chromatography (Shimadzu, Model: QP2010S, Tokyo, Japan) to analyze its chemical composition. A gas chromatography column (ELITE-5MS) (internal diameter 0.25 mm, thickness 0.25 µm, 30 m length) was coupled to a mass detector (with 5% diphenyl and 95% 140 dimethylpolysiloxane) using standard GC-MS parameters. The carrier gas was helium maintained at a column flow of 1.0 mL/min (at a pressure of 53.5 kPa). Then, 1 μL of sample from 1 mg/mL stocks was injected by split injection mode into the system having an injection temperature of 260 °C. The initial temperature was set at 80 °C for 3 min, which was programmed to increase to 280 °C at a rate of 5 °C/min. The material was analyzed using the Mass Spectrometric Detector (MSD) in ACQ Scan mode with an m/z range of 40–600. Automatic screening software was used to screen volatile and semi-volatile chemicals. The significant peaks were compared with the M.S. reference database of NIST 11 (National Institute of Standards and Technology, Gaithersburg, MD, USA) and WILEY 8.

The same experimental setup was used to inject a mixture of linear hydrocarbons (C9 to C25 alkanes). The constituents in the essential oil were confirmed by comparing the observed mass spectra with those from the equipment database (Wiley 8 lib and NIST 11 lib) and by utilizing the Retention Index (RRI), calculated for each ingredient, as previously described [15].

### 4.4. Linalool 

The analyte Linalool, showing a prominent peak in GC-MS spectra, was procured from Sigma Aldrich, and a 1% stock solution using ethanol was prepared and stored at 4 °C for further use.

### 4.5. Determination of Minimum Inhibitory Concentrations

The Minimum Inhibitory Concentration (MIC) of Linalool against *S. pyogenes* was determined by the broth microdilution method, using THYG broth in 96-well microtiter plates (Thermo Scientific Nunc, Waltham, MA, USA) following the CLSI guidelines [44]. To the 96-well microtiter plates containing 196 µL of THYG broth, 2 µL of Linalool was added in increasing concentrations. The concentration of the solvent was maintained at 1% in all wells. A test pathogen at 1% concentration was added [25]. The broth with solvent was kept blank, and broth with the organism alone was the control. The microtiter plates were incubated at 37 °C for 16–18 h, and the O.D. was measured at 600 nm using a multi-plate reader (Tecan Spark10M, Zurich, Switzerland). The lowest concentration inhibiting visible bacterial growth, similar to that of the blank, was taken as the MIC [42]. The Resazurin Microtiter assay was used for confirmation of MIC. All the steps followed were the same as above until incubation. After incubation, 20 µL of Resazurin solution (33.75 mg/5 mL) was added to all the wells, and the plates were wrapped with aluminum foil and kept for 4–6 h for incubation. The color shift was then visually evaluated. Resazurin reduction caused by bacterial development was indicated by a shift from blue to pink. The lowest concentration that prevented the color change was taken as the MIC [45].

### 4.6. Determination of Minimum Biofilm Inhibition Concentration (MBIC)

The minimum concentration that showed maximum reduction in biofilm formation compared to the control was considered the MBIC. The microdilution method determined the MBIC of Linalool against *S. pyogenes* in 24-well polystyrene microtiter plates with 1 mL of THYG broth. Linalool at an increasing concentration (0.0002% to 0.006%) was added to 1 mL THYG broth containing 1% biofilm-forming *S. pyogenes* cell suspension. The well-inoculated cells alone and uninoculated sterile broth with solvent alone served as the control and blank, respectively. The final solvent volume in each well was made up to 1%. The plates were then incubated at 37 °C for 24 h, and to evaluate the antibacterial activity, the cell density was read at 600 nm. The planktonic cells and loosely bound cells were washed off with distilled water, the plates were allowed to dry, and the biofilm formed was stained with 1 mL of 0.4% (*w*/*v*) crystal violet (CV) (Hi-Media, Mumbai, India) for 10 min. To remove unbound stains, the wells were washed twice with distilled water and dried. The biofilm bound to crystal violet was extracted using glacial acetic acid (20%) for 10 min and read at 570 nm (Tecan Spark 10M, Zurich, Switzerland). The amount of crystal violet stain recovered from the wells was proportional to the cells in the biofilm. Hence, lower absorbance at 570 nm in the treated wells indicates biofilm inhibition. 

The percentage inhibition was measured as % of inhibition = ((Control OD_570_ − Treated OD_570_)/Control OD_570_) × 100 [46].

### 4.7. Colony-Forming Unit (CFU) Analyses

*S. pyogenes* were grown in THYG broth, with and without Linalool (at MBIC), for 24 h. The control and treated cultures were serially suspended in 1 × PBS and spread on THYG agar plates. Plates were incubated at 37 °C for 24 h to count CFU [25].

### 4.8. Cell Viability Assay 

The cytotoxic effects of Linalool at its MBIC against *S. pyogenes* were analyzed using the AlamarBlue^TM^ cell proliferation assay kit (Bio-Rad, Berkeley, CA, USA). *S. pyogenes* was cultured in 1 mL THYG broth with and without Linalool and incubated at 37 °C for 24 h. The cell pellet was collected and resuspended in 1 mL of 0.9% saline, and then 100 mL of AlamarBlue^TM^ (1 mg/mL) was added. For 4–6 h, multi-well plates were incubated in the dark. O.D. was measured at 570 and 600 nm. The percentage difference in the reduction of AlamarBlue^TM^ was determined according to the manufacturer’s instructions between treated and control samples [7].

### 4.9. In Situ Visualization of Biofilm Inhibition

For microscopic visualization, *S. pyogenes* cells in the presence and absence of Linalool (0.004%) were allowed to develop biofilm for 24 h at 37 °C on 1 × 1 cm glass slides and placed in 24-well plates with 1 ml of THYG broth. The glass slides were then washed with PBS and air-dried. Based on the microscopic analysis, the slides were stained and fixed accordingly [39].

For light microscopic visualization, glass slides were stained with 0.4% crystal violet and incubated for 10 min. Glass slides were washed with distilled water to remove excess stain and air dried. The air-dried slides were viewed under a light microscope (Olympus: C×43, Tokyo, Japan) at a magnification of 400×, and the images were documented with an accompanying digital camera (Magcam, model: DC 10, New Delhi, India) [25].

For confocal laser scanning microscopic (CLSM) analysis, glass slides were stained with 0.1% acridine orange and incubated for 5 min in the dark. The slides were washed with distilled water to remove excess stains. Subsequently, the slides were air-dried and viewed under CLSM (LSM 710, Carl Zeiss, Oberkochen, Germany) [25].

For scanning electron microscopic (SEM) analysis, the biofilm formed on the glass slides was fixed with 2.5% glutaraldehyde for 8 h at 4 °C, followed by dehydration with increasing concentrations of ethanol (20%, 40%, 60%, 80%, and 100%) for 5 min each. Eventually, the slides were air-dried thoroughly and gold-coated before observation under SEM (Jeol-JSM 5600LV, Tokyo, Japan) [28].

### 4.10. Cytotoxic Assay 

The cytotoxicity of Linalool against Human Embryonic Kidney (HEK) cell lines was measured using a modified version of Mosmann’s 3-(4,5- dimethylthiazol-2-yl)-2, 5-diphenyltetrazolium bromide (MTT) reduction test [47]. Cells were seeded at 2500 cells/well in 96-well microtiter plates and allowed to acclimate to the culture conditions, such as 37 °C and 5% CO_2_ environment, in the incubator for 24 h. Linalool was prepared in DMEM medium (100 mg/mL), sterilized with a Millipore syringe filter (0.2 m), and added to the wells containing cells. Untreated wells were kept in the control. The plates were then incubated for 24 h after being treated with the test samples. The media from each well were aspirated and washed with Phosphate Buffered Saline (PBS). One hundred microliters of fresh media was added to each well, and 100 μL of 0.5 mg/mL MTT solution in PBS was added to each well. The plates were then incubated for an additional 2 h to form formazan crystals. After removing the supernatant, 100 μL of 100% DMSO was added to each well to solubilize the formazan crystals. The absorbance at 570 nm was measured with a microplate reader (Tecan Spark 10M, Zürich, Switzerland). Two wells per plate without cells served as blanks. Cell viability was expressed using the following formula:(1)$$\mathrm{Percentage}\;\mathrm{of}\;\mathrm{cell}\;\mathrm{viability}=\frac{\mathrm{Average}\;\mathrm{absorbance}\;\mathrm{of}\;\mathrm{treated}}{\mathrm{Average}\;\mathrm{absorbance}\;\mathrm{of}\;\mathrm{control}}\times 100$$

### 4.11. Microbial Adhesion to Hydrocarbon (MATH) Assay 

The effect of Linalool on the cell surface hydrophobicity of *S. pyogenes* was investigated using the MATH assay, which assesses a cell’s capacity to cling to a hydrophobic substrate. *S. pyogenes* cultures were grown in the presence and absence of Linalool (0.004%) for 24 h at 37 °C in THYG broth. The cultures were diluted to obtain an O.D. of 0.4 at 600 nm. One ml of toluene was added to an equal volume of diluted culture and vortexed thoroughly for 2 min. Until the phases separated, the tubes were left undisturbed at room temperature. After separation, the aqueous phase O.D. was measured at 600 nm. 

Cell surface hydrophobicity was estimated using the formula: [(OD_600nm_ before vortexing − OD_600nm_ after vortexing)/OD_600nm_ before vortexing] × 100 [48,49]

### 4.12. Extracellular Polymeric Substance (EPS) Quantification

The effect of Linalool on EPS generation was assessed by measuring the total carbohydrates around the bacterial cells. *S. pyogenes* cultures were grown for 24 h at 37 °C in the presence and absence of Linalool (0.004%). The cultures were centrifuged for 10 min at 10,000 rpm, and the pellets were washed three times in sterile PBS before being resuspended in 200 µL. To the cell suspension, an equal volume of 5% phenol and five volumes of concentrated sulfuric acid containing 0.2% hydrazine sulfate were added, mixed, and incubated at room temperature for 1 h in the dark. The samples were centrifuged for 10 min at 10,000 rpm, and the absorbances of the supernatants were measured at 490 nm [48].

### 4.13. Secreted Protease Quantification

An Azocasein assay was done to quantify the total cysteine protease production [48]. *S. pyogenes* was cultured for 24 h at 37 °C in the absence and presence of Linalool (0.004%). The supernatants were sterilized using a 0.2-micron nylon membrane filter after the culture was centrifuged at 12,000 rpm. The cell-free culture supernatant was mixed with an equal volume of activation buffer and maintained at 40 °C for 30 min. An equivalent volume of 1% (*w*/*v*) azocasein was added to the mixture and incubated for another 1 h at 40 °C. To precipitate the protein and terminate the process, trichloroacetic acid (10%) was added and mixed thoroughly. After centrifuging the mixture for 5 min at 12,000 rpm, 600 μL of supernatant was transferred to new tubes. To this, 700 μL of 1 N NaOH was added, and their absorbance was measured at 440 nm [26]. The change in absorbance value was expressed as the activity. 

### 4.14. Real-Time PCR Analysis

Total RNA from overnight grown *S. pyogenes* SF370 in the absence and presence of Linalool (0.004%) for 24 h was isolated according to the method described by Oh and So [50]. A Thermo Verso cDNA synthesis kit (Thermoscientific, Waltham, MA USA) was used to generate cDNA according to the manufacturer’s instructions. Real-time PCR (CFX Connect, Bio-Rad, Berkeley, CA, USA.) was performed with gene-specific primers, as listed in Table 2, using the SYBR green kit (Applied Biosystems, Waltham, MA USA). Gyrase (*gyr*) was used as the housekeeping gene for internal control. The PCR cycle consisted of an initial denaturation at 94 °C for 5 min, followed by a 30-cycle amplification consisting of denaturation at 94 °C for 45 sec, annealing at 58.5 °C for 45 s, and an extension at 72 °C for 30 s, followed by final extension at 72 °C for 10 min. The expression patterns of candidate genes were normalized against *gyr* expression (housekeeping gene) and quantified using the 2^−ΔΔCT^ method [51].

### 4.15. Histopathology Analysis Using Danio Rerio

All experiments in *Danio rerio* were performed following the general guidelines of the Institutional Animal Ethics Committee, Alagappa University (IAEC/AU/OCT 2021/Fish-1). The toxicity of Linalool, if any, and its histopathological changes were evaluated *in vivo* using a *D. rerio* survival assay. The toxicity of Linalool was assessed by ascertaining the survival percentage of Zebrafish grown in the absence and presence of Linalool at various concentrations (MBIC × 4, MBIC × 2, MBIC, MBIC/2). Fish behavior was evaluated visually after exposure to the compound at different concentrations to spot any potential anomalous behavior. Histopathological analysis was also performed to analyze the effects of Linalool on the significant tissues. Treated fish were carefully euthanized and fixed with 10% (*v*/*v*) phosphate-buffered formalin, containing 0.4% NaH_2_PO_4_, 0.65% Na_2_HPO_4_, and 40% formalin. The tissues (gills, kidney, and liver) were sectioned transversely and stained with Hematoxylin–Eosin (HE) (Bio-Optica, Milano, Italy). The samples were examined using a light microscope (Leica DM750) to identify potential morphological alterations and were documented in Nikon Eclipse, Ti 100 [40].

### 4.16. Statistical Analysis 

All experiments were performed in biological triplicate with three technical replicates. Statistical analyses were performed using Prism software (version 8.0.2 (263), GraphPad Software Inc., La Jolla, CA, USA). The values are expressed as a mean ± standard deviation (S.D.). One-way ANOVA and Student’s *t*-test were used to compare the treated and control groups. A *p*-value < 0.05 was considered statistically significant. 

## 5. Conclusions

The present study explored Linalool’s anti-biofilm and anti-virulence potential, a major compound from the essential oil from the rhizomes of *Hedychium larsenii* against *S. pyogenes*. At 0.004%, Linalool exhibited potential anti-biofilm activity without exerting any pressure on the growth and metabolic activity of *S. pyogenes*. Light and confocal microscopy revealed that Linalool impaired biofilm architecture and reduced microcolonies in the treated sample compared to the control. Linalool decreased cell surface hydrophobicity and extracellular polymeric substance production. Differential regulation of *mga, speB, srv, dltA,* and *ropB* makes it clear that Linalool inhibits biofilm formation in *S. pyogenes* by lowering CSH and impairing the biofilm matrix. Essential virulence factor-coding genes, such as *hasA, slo*, and *col370*, were downregulated, which suggested reduced pathogenicity. The non-toxic nature of Linalool in HEK cell lines and, as evident from the histopathological analysis in *D. rerio*, makes Linalool a promising therapeutic agent in treating biofilms formed by *S. pyogenes.*

## Figures and Tables

**Figure 1 antibiotics-12-00545-f001:**
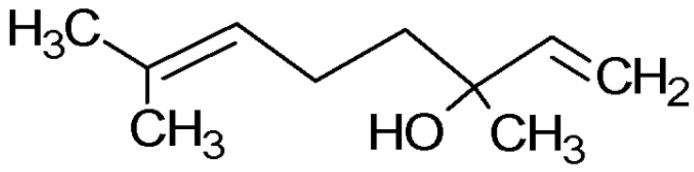
Structure of Linalool.

**Figure 2 antibiotics-12-00545-f002:**
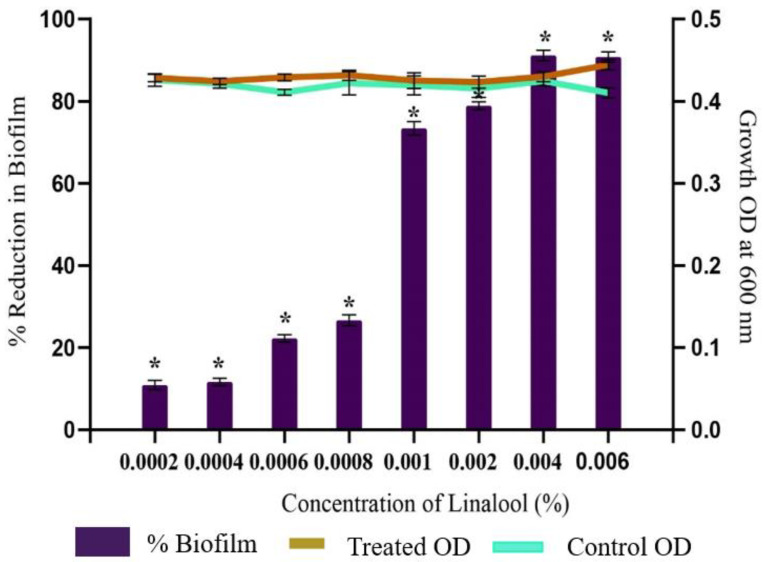
Effect of increasing the concentration of Linalool on the growth and biofilm development in *S. pyogenes*. Graph displaying anti-biofilm activity of Linalool at different concentrations (0.0002% to 0.006%) against *S. pyogenes* SF370. A concentration of 0.004% was considered the MBIC exhibiting 91% inhibition of biofilm without affecting growth. Error bars represent the S.D. Asterisks indicate statistical significance (*p* < 0.05).

**Figure 3 antibiotics-12-00545-f003:**
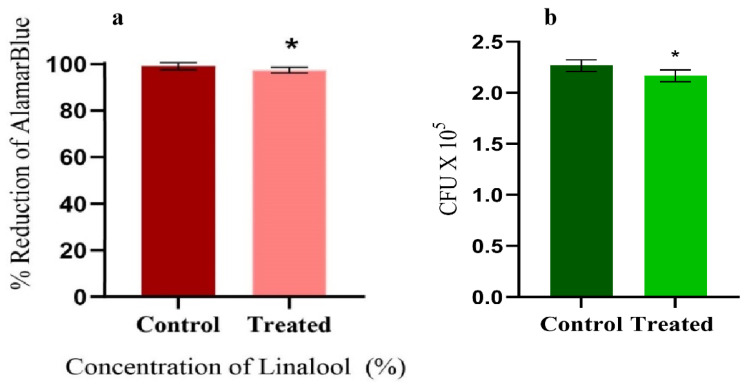
(**a**) AlamarBlue^TM^ assay showing the metabolic viability of the control and Linalool-treated *S. pyogenes* at its MBIC (0.004%). (**b**) CFU analysis of the control and Linalool-treated *S. pyogenes* exhibiting the non-antibacterial nature of Linalool at its MBIC (0.004%). Error bars represent the S.D. Asterisks indicate statistical significance (*p* < 0.05).

**Figure 4 antibiotics-12-00545-f004:**
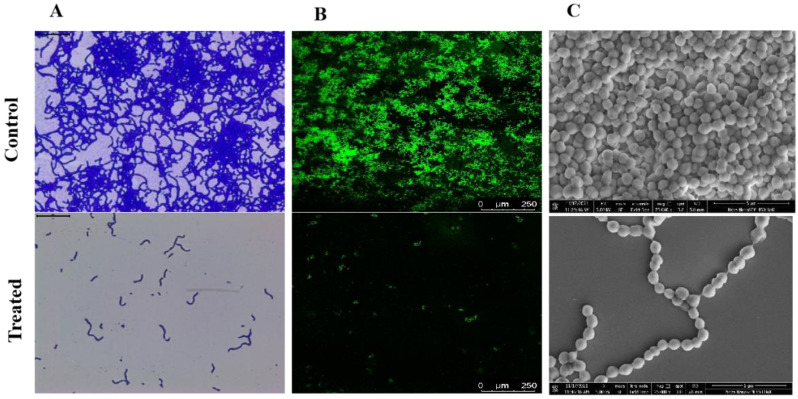
Microscopic visualization of *Streptococcus pyogenes* biofilm. (**A**) Light microscopic images (400×) representing Linalool’s inhibitory effect on biofilm microcolony formation. (**B**) CLSM images (200×) of the acridine orange-stained biofilm of *S. pyogenes*, showing a decrease in biofilm architecture formation on Linalool treatment. Scale bar 250 μm. (**C**) SEM images (25,000×) comparing the cell surface features of *S. pyogenes* biofilms produced in the presence and absence of Linalool (at its MBIC of 0.004%). Scale bars 5 μm.

**Figure 5 antibiotics-12-00545-f005:**
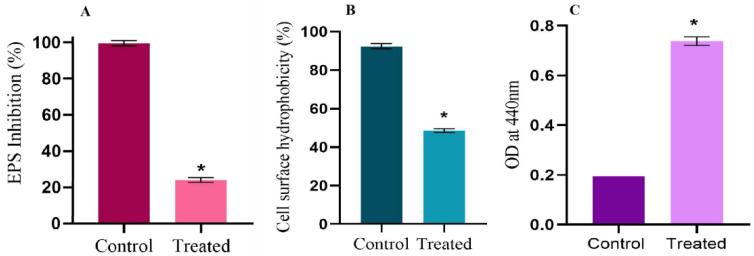
Effect of Linalool on EPS production, cell surface hydrophobicity, and extracellular protease production. (**A**) Percentage inhibition of EPS grown in the presence of Linalool (0.004%). (**B**) Graph showing the cell surface hydrophobicity (CSH) of *Streptococcus pyogenes* SF370 in the absence and presence of Linalool (0.004%). (**C**) The graph displays extracellular cysteine protease production by *S. pyogenes* in the absence and presence of Linalool at the MBIC concentration. Data are represented as mean ± S.D. Asterisks denote statistical significance (*p* < 0.05).

**Figure 6 antibiotics-12-00545-f006:**
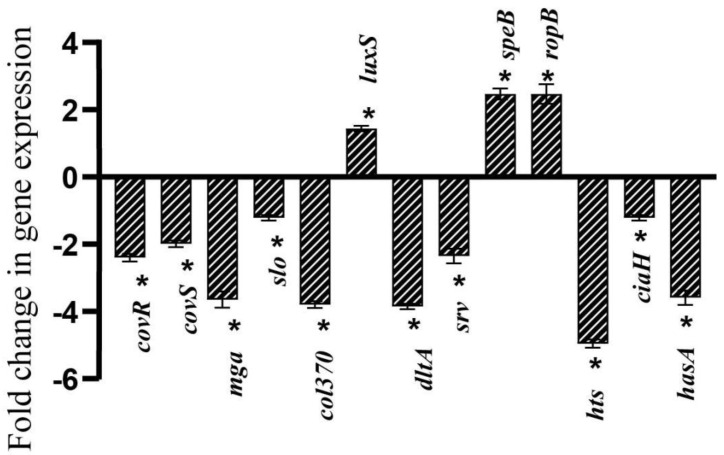
Linalool modulates virulence gene expression differentially. qPCR analysis of *S. pyogenes*’ expression of biofilm- and virulence-related genes following a 24-h Linalool treatment. Data are represented as S.D. Asterisks denote statistical significance (*p* < 0.05).

**Figure 7 antibiotics-12-00545-f007:**
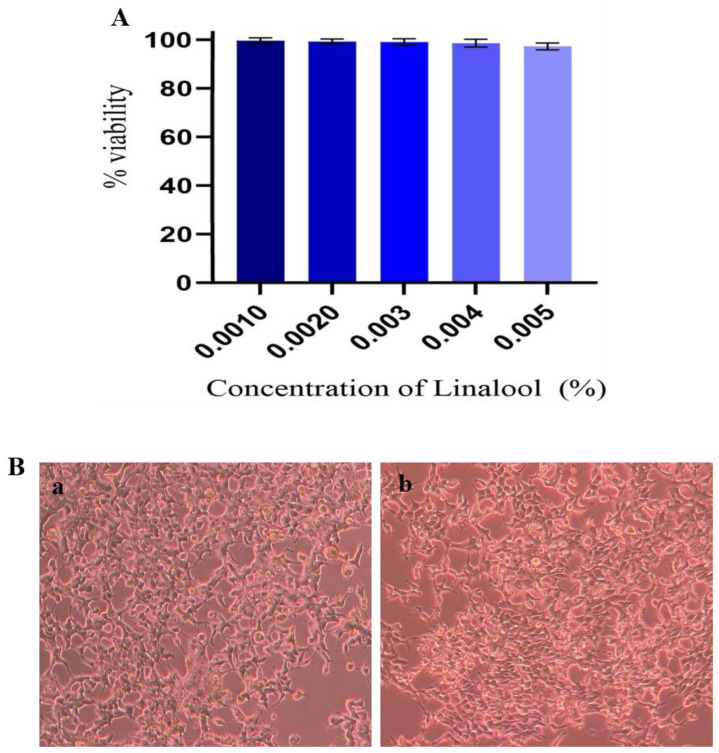
(**A**) Effect of Linalool on HEK cells. HEK cells treated with different concentrations of Linalool showed 100% viability at the tested concentrations. Data are represented as S.D. (**B**) Effect of Linalool on HEK (Human Embryonic Kidney) cells. (**a**) Control HEK cells (**b**) HEK cells treated with Linalool (0.004%) were healthy and normal compared to the control.

**Figure 8 antibiotics-12-00545-f008:**
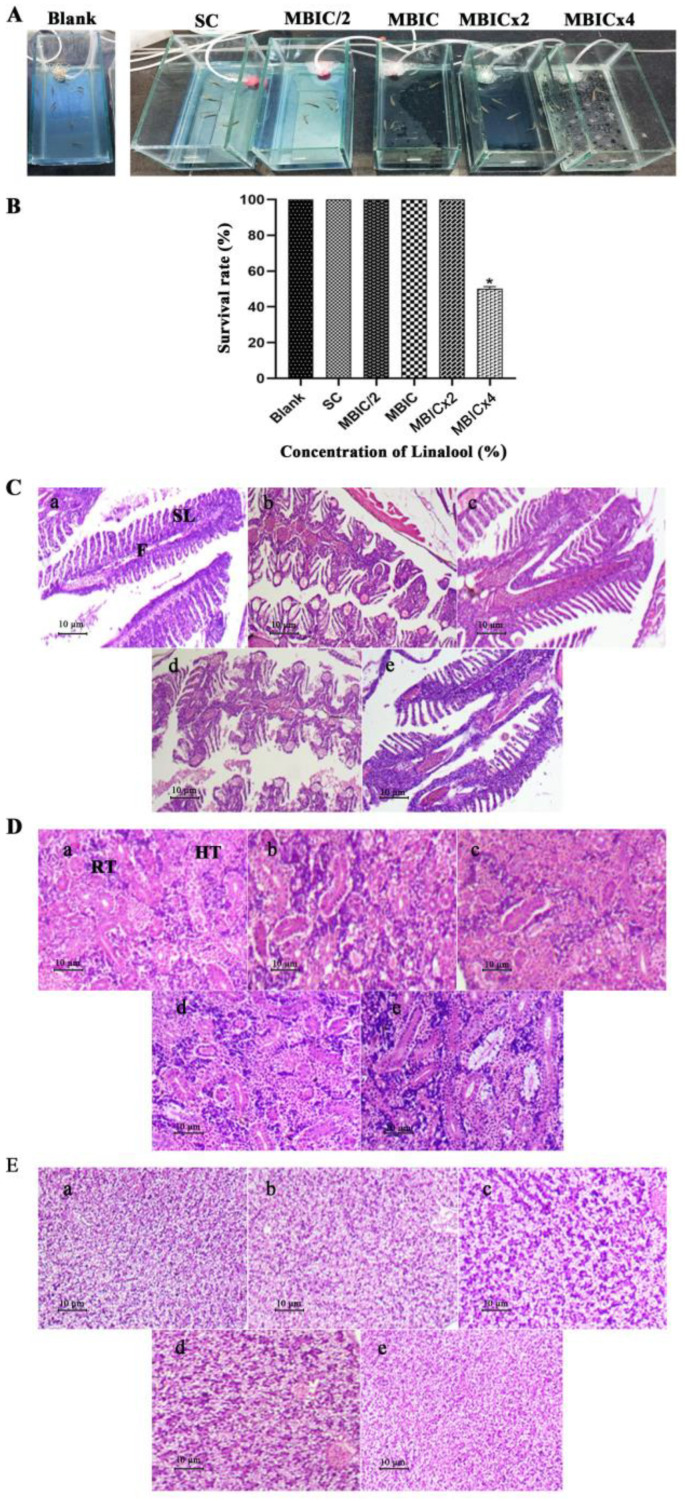
(**A**) Experimental setup for establishing an *in vivo* toxicity analysis on *Danio rerio* at different concentrations of Linalool. S.C. represents the solvent control. (**B**) *In vivo* study in *Danio rerio***.** Survival percentage of Zebrafish at different concentrations of Linalool (%). Data are expressed as the mean ± S.D., *p* < 0.05. Blank is the control group, and S.C. is the solvent control. Linalool-treated histopathological samples of vital organs after the 4th day of treatment. (**C**) Light micrographs of gill morphology of adult zebrafish exposed to Linalool for 96 h (100×). (**a**) Control group; (**b**) solvent control exhibiting normal structure, including a gill arch formed by filaments (F) and lamellae (S.L.); (**c**–**e**) gills exposed to MBIC/2, MBIC, and MBIC × 2 of Linalool exhibited normal histology. (**D**) Light micrographs of the kidneys of adult zebrafish exposed to Linalool for 96 h (100×). (**a**) Control group; (**b**) solvent control exhibiting normal histology with regular renal tubules (R.T.); (**c**–**e**) kidney from fish exposed to MBIC/2, MBIC, MBIC × 2 exhibiting normal histology. (**E**) Light micrographs of liver of adult zebrafish exposed to Linalool for 96 h (100×). (**a**) Control group (**b**) solvent control exhibiting normal structure with normal hepatocytes (H); (**c**–**e**) liver from fish exposed to MBIC/2, MBIC, and MBIC × 2 exhibited normal histology.

**Table 1 antibiotics-12-00545-t001:** List of major phytochemicals identified through GC-MS analysis.

Peak No.	R. Time	Area%	Name of the Metabolites	RRI (Calculated)	RRI (Reference)
1	5.221	1.02	α-Pinene	932	932
2	5.639	0.22	Camphene	946	946
3	6.383	3.72	β-Pinene	976	974
4	6.654	0.20	Myrcene	987	988
5	7.794	11.77	Cymene	1023	1022
6	7.952	1.00	Limonene	1027	1024
7	8.069	14.48	Eucalyptol, 1.8-Cineole	1025	1026
8	8.933	0.24	1.4-cyclohexadiene,1-methyl-4-(1- methylethyl)-	1055	1054
9	9.390	2.87	linalool oxide B	1068	1067
10	9.981	2.81	Trans-linalool oxide	1085	1084
11	10.592	52.11	Linalool	1102	1095
12	11.224	0.25	α-fenchol	1117	1114
13	12.071	0.15	Pinocarveol	1138	1135
14	13.456	0.32	Linalool oxide trans-pyranoid	1172	1173
15	13.692	1.38	Terpinen-4-ol	1177	1174
16	14.298	2.95	α-Terpineol	1192	1186
17	18.069	0.29	Isobornyl acetate	1280	1283
18	18.348	0.20	Thymol	1287	1289
19	18.683	0.26	Carvacrol, ethyl ether	1295	1297
20	26.337	0.30	Selina-3,7(11)-diene	1540	1545
21	27.730	0.29	α-selinene	1512	1520
22	33.230	0.31	Pogostol	1651	1651

R. Time refers to the compound’s retention time as obtained through GCMS analysis. RRI (calculated) represents the relative retention index, and RRI (reference) indicates the R. Indices in the Adams mass spectral library.

**Table 2 antibiotics-12-00545-t002:** List of genes used for real-time PCR analysis, with their functions and primer sequences [48,52,53].

Gene	Function	Primer Sequence (5′-3′)	
		Forward	Reverse
*mga*	Virulence factor regulation and Biofilm formation.	GATCCGTTACTACAAGGG	GTTACTTGTCTGCCTCCT
*ompA*	Outer membrane protein involved in stress response	GTGCTTCCTGGCTATGAACC	GCAGCGGGTTGGTTATTGTA
*covR*	Repressor gene of TCS of covRS. Stress response, biofilm formation, and regulate 15% of genes.	TGCGCGTGATTCTATTATGG	GGCGGAAAATAGCACGAATA
*sagA*	Streptolysin S production	AAACAACTCAAGTTGCTCCTG	TGGCGTATAACTTCCGCTAC
*covS*	TCS, control of virulence sensor, Regulates biofilms	GAGTGAGCGCGATATCACAA	GCAAGCCAGGAGATGATTCT
*hlyX*	Hemolysin production	GCGCAATACCCAAAATCAGA	CGATTTCACCGACGATTTCT
*slo*	Streptolysin O synthesis	GCCAATGTTTCAACAGCTATTG	CGGAGCTGCACTAAAGGCCGC
*col370*	Involved in adhesion	AACCCAGATACTGCACCACA	GCGAGCTGATTACCACCTTG
*dltA*	D-alanylation of LTA	GCATTTGGACATCGACTCCT	GTTTTCGAGCCGTAGAAACG
*htsA*	Heme-transporter gene	ATTGTAGCCACTTCGGTTGC	AAACCCACACGCTTAACAGC
*srv*	Regulation of virulence factor and Biofilm formation	CGGCATTGTGAAACAGAGTG	TCTGACTCGATGCGAACATT
*speB*	Production of extracellular cysteine protease	CTAGGATACTCTACCAGCG	CAGTAGCAACACATCCTG
*hasA*	Production of Hyaluronic acid capsule.	AGCGTGCTGCTCAATCATTA	AACATCGATCATCCCCAATG
*ropB*	Transcriptional factor. Regulates virulence and stress.	TGATATGGATACGGCAAAACA	TTGACCAAGGCAAAAAGGTT
*luxS*	Virulence factor regulation	CTTTTGGCTGTCGAACAGGT	TCCAGGAACATCTTCCCAAG
*spy125*	Synthesizes minor pilin subunits.	AGAGATTAGCGACGCAACAG	ATGGCCATATGTCTCCACCA
*srtB*	Production of Class C sortase, involved in aggregation	GCTGGTTTTGGTTTGTGGGA	CCCCGGGATATTTAACCAACC
*ciaH*	Stress response TCS	GGCGGTCTTACAGAATCGTC	CATGTTGCGAACCTCGTCTA
*gyrA*	Gyrase production (House-keeping gene in the present study)	CAACGCACGTAAGGAAGAAA	CGCTTGTCAAAACGACGTTA

## Data Availability

Not Applicable.

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
