# Peer review of "The Anti-Biofilm Potential of Linalool, a Major Compound from *Hedychium larsenii*, against *Streptococcus pyogenes* and Its Toxicity Assessment in *Danio rerio"

_antibiotics, 2023, doi:10.3390/antibiotics12030545_

Round 1

Reviewer 1 Report

The manuscript  found and explain the action of Linalool alcohol  on Streptococcus pyogenes biofilm formation, the way all the experiments were performed and demonstrated was more convincing.

1. It would be more useful if the author include the compound molecular structure.

2. The same title compound was  reported as antibacterial and significant anti biofilm actions, it would be more useful for readers if the author could be  include those information (1. https://www.ncbi.nlm.nih.gov/pmc/articles/PMC6938037/. 2.https://www.frontiersin.org/articles/10.3389/fmicb.2019.02947/full). 3.https://www.ncbi.nlm.nih.gov/pmc/articles/PMC7796449/) etc..

3. The provided Scanning electron microscope images in Fig.3 with and without sample treatment are not prominent evidencing, author can follow or site (https://www.mdpi.com/1420-3049/27/4/1167).

Author Response

Comments to the Author

It would be more useful if the author include the compound molecular structure

Ans:  The molecular structure of the compound Linalool is included in the introduction section as Figure 1. The subsequent figures' numbering changed and thus was highlighted.

Comment: The exact title compound was reported as antibacterial and significant anti biofilm actions, it would be more useful for readers if the author could include those information (1. https://www.ncbi.nlm.nih.gov/pmc/articles/PMC6938037/. 2.https://www.frontiersin.org/articles/10.3389/fmicb.2019.02947/full). 3.https://www.ncbi.nlm.nih.gov/pmc/articles/PMC7796449/) etc..

Ans: The references suggested have been included in the Introduction and Discussion session (changes highlighted in yellow color).

Comment: The provided Scanning electron microscope images in Fig.3 with and without sample treatment are not prominent evidencing, author can follow or site (https://www.mdpi.com/1420-3049/27/4/1167).

Ans: The SEM images have been changed with better resolution in Figure 4 c. The Scanning electron microscope image of the untreated sample showed an aggregated form of bacterial chains with sticky EPS encircling the cells. But the treated sample had a clear cell surface and dispersed chains of the bacterial cell, indicating decreased EPS production. This demonstrates an apparent decrease in biofilm in the treated sample.

Reviewer 2 Report

In this manuscript, the authors investigated the anti-biofilm activity of Linalool against Streptococcus pyogenes and its toxicity assessment in Denio rerio. Here are some comments.

General comments

1. The authors extracted essential oil from Hedychium larsenii using hydrodistillation and then the essential oil was subjected to phytochemical analysis using GC-MS. I thought the authors wanted to use the extracted essential oil for the next experiments, but Linalool used in this study was purchased from company. What is the purpose to extract and analyze essential oil in this case? It makes no sense to describe 4.2 and 4.3 and these sections should be deleted.

2. The authors emphasized the Linalool was from Hedychium larsenii in title. In fact, the Linalool used in this study was purchased from company, but not extract from Hedychium larsenii by yourselves.

3. Line 107-110: optical density did not reflect cell viability. If the authors wanted to prove the biofilm suppression was not due to growth inhibition, plate counting is better.

4. The discussion section is too long. Results should not be mentioned in discussion. The citation of figures in discussion should be avoided.

5. The format of the reference is very confusing. Please standardize the format of the reference according to the requirements of the journal.

Author Response

Thanks for the valuable suggestions. 

Comments to the Author

The authors extracted essential oil from Hedychium larsenii using hydrodistillation and then the essential oil was subjected to phytochemical analysis using GC-MS. I thought the authors wanted to use the extracted essential oil for the next experiments, but Linalool used in this study was purchased from company. What is the purpose to extract and analyze essential oil in this case? It makes no sense to describe 4.2 and 4.3 and these sections should be deleted

Ans:  We have extracted the essential oil from Hedychium larsenii and conducted preliminary experiments to test its efficacy as an anti-biofilm agent. After proving the anti-biofilm activity, the essential oil was subjected GC-MS analysis. The prominent peak in GC-MS spectra was identified as 'Linalool' with an area of 52.11%. Being the primary compound, we bought the product from Sigma and conducted further detailed experiments. The results demonstrated exceptional biofilm inhibition by Linalool at the lowest concentration of 0.004%, then the crude oil itself. We also screened other compounds for anti-biofilm activity using a crystal violet assay. But we found that Linalool showed the best activity at the lowest concentration. As the significant compound, we narrowed it to Linalool based on other in-vitro studies. Sections 4.2 and 4.3 shows how we narrowed down to Linalool from the various compounds obtained from GC-MS results. In this study, we followed the theme of the published papers mentioned below, which employs a similar method of identifying compounds without moving into the purification of the candidate compound.

Lakshmi, S.A.; Bhaskar, J.P.; Krishnan, V.; Sethupathy, S.; Pandipriya, S.; Aruni, W.; Pandian, S.K. 2020. Inhibition of biofilm and biofilm-associated virulence factor production in methicillin-resistant Staphylococcus aureus by docosanol. Journal of biotechnology317, pp.59-69.

Prasath, K.G.; Sethupathy, S.; Pandian, S.K. 2019. Proteomic analysis uncovers the modulation of ergosterol, sphingolipid and oxidative stress pathway by myristic acid impeding biofilm and virulence in Candida albicansJournal of proteomics208, p.103503.

Comment: The authors emphasized the Linalool was from Hedychium larsenii in title. In fact, the Linalool used in this study was purchased from company, but not extract from Hedychium larsenii by yourselves.

Ans: Linanool is the primary compound identified by GC-MS analysis of the essential oil of H. larsenii. Once the major molecule in the active fraction was identified, we proceeded with detailed experiments with pure Linalool purchased from Sigma. If the title is confusing, it may be changed to "The anti-biofilm potential of Linalool, a major compound from Hedychium larsenii against Streptococcus pyogenes and its toxicity assessment in Danio rerio.

Comment: Line 107-110: optical density did not reflect cell viability. If the authors wanted to prove the biofilm suppression was not due to growth inhibition, plate counting is better.

Ans: Thank you for the valuable comments. To assess the effect of Linalool on the growth and metabolism of S. pyogenes, OD at 600nm at various tested concentrations and cell viability assays were done. Considering your suggestion, we also confirmed the results through CFU analysis. No reduction was observed in OD at 600nm (Figure 2) or CFU values of Linalool-treated cells compared to control cells (Figure 3b). AlamarBlueTM assay is the gold standard technique, used for more than five decades to assess the metabolic viability of cells. No significant change in the percentage reduction of AlamarBlueTM between Linalool-treated and control cells was seen. Compared to control samples, the reduction in Linalool treated sample was found to be 97.38% (Figure 3a) (changes highlighted in yellow).

The following articles were considered for evaluating the results.

 Lakshmi, S.A., Bhaskar, J.P., Krishnan, V., Sethupathy, S., Pandipriya, S., Aruni, W. and Pandian, S.K., 2020. Inhibition of biofilm and biofilm-associated virulence factor production in methicillin-resistant Staphylococcus aureus by docosanol. Journal of biotechnology317, pp.59-69

Valliammai, A., Selvaraj, A., Sangeetha, M., Sethupathy, S. and Pandian, S.K., 2020. 5-Dodecanolide inhibits biofilm formation and virulence of Streptococcus pyogenes by suppressing core regulons of virulence. Life Sciences262, p.118554.)

Comment: The discussion section is too long. Results should not be mentioned in discussion. The citation of figures in discussion should be avoided.

Ans: The discussion section has been made short and the citation of figure is removed from the discussion section.Changes highlighted.

Comment: The format of the reference is very confusing. Please standardize the format of the reference according to the requirements of the journal.

Ans: We have modified the format of the references and corrected the references as per the requirements of the journal.Changes highlighted

Reviewer 3 Report

The studies carried out are relevant and have important scientific and practical significance for preventing the formation of microbial biofilms in the medical and industrial fields. However, while reading, some questions and comments arose:

1. Figure 3. The presented microphotographs do not have a uniform design style. Somewhere the magnification of the microscope is indicated, and somewhere - the scale bar. It would be better to put a scale bar on microphotographs.

2. Figure 3C. On the presented SEM microphotographs, it is necessary to show where the EPS is located.

3. Figure 4C. What is the reason for the increase in extracellular protease in your case?Why is the protease activity presented in studies in the form of optical density data, and not in percentages?

4. Figure 7. It is better to indicate the scale bar on microphotographs. The letter designations are incomprehensible, it is better to highlight these areas somehow.

5. Discussions. In the description of histostreses of the liver, kidneys and gills, you speak of the absence of obvious histopathological changes. So there were some histological changes compared to control samples?

6. DOIs for articles are not indicated in the list of references.

Author Response

Thanks for valuable comments.

Comments to the Author

Figure 3. The presented microphotographs do not have a uniform design style. Somewhere the magnification of the microscope is indicated, and somewhere - the scale bar. It would be better to put a scale bar on microphotographs.

Ans: Magnification of microscope and scale bars have been included and made uniform. Changes highlighted in the figure caption, Figure 4.

Figure 3C. On the presented SEM microphotographs, it is necessary to show where the EPS is located.

Ans: The control cells are embedded in the EPS, whereas the treated cells are showing clear morphology. SEM images of control surface were completely covered with chains of Streptococcal cells. Multi-layer biofilm with microcolony formation was also observed. In case of Linalool treatment, biofilm formation was almost completely arrested and individually dispersed cells were observed

Figure 4C. What is the reason for the increase in extracellular protease in your case? Why is the protease activity presented in studies in the form of optical density data, and not in percentages?

Ans: Linalool treatment enhanced extracellular protease production.  Although S. pyogenes secrete multiple proteases, Streptococcal pyrogenic exotoxin B (SpeB) is the predominant extracellular protease. It plays a key role in virulence by degrading immune modulators and host matrix proteins, and facilitates host tissue damage and dissemination. Interestingly, SpeB also degrades the peptides that stabilize the biofilm matrix and thus regulates biofilm formation and dispersal. Hence, the enhanced protease production observed upon Linalool treatment would deter biofilm formation by proteolysis. To further assess this hypothesis, real-time PCR was performed, which showed the upregulation of speB by 2.4-fold. The expression of speB is regulated by the independent stand-alone regulators Srv and RopB. speB is positively regulated by RopB, but speB is negatively regulated by Srv. Linalool increased the expression of ropB while downregulating srv. Thus, increased expression of ropB led to higher levels of expression of speB, as evident from RT-PCR studies. And decreased level of expression by srv, eventually lead to increased level of speB production. The combined effects of decreased Cell surface hydrophobicity (CSH), elevated protease and RT-PCR analysis show that numerous gene regulation pathways are the main cause of decreased biofilm formation after Linalool treatment.

Protease activity can be showed in OD values, as well as in percentage inhibition. Compared to control samples, if the OD values are decreasing, we can say that its protease inhibition or vice-versa. In this study we followed the method of below mentioned published papers which employs a similar method of identifying protease activity by measuring OD values.

Nandu, T.G., Subramenium, G.A., Shiburaj, S., Viszwapriya, D., Iyer, P.M., Balamurugan, K., Rameshkumar, K.B. and Karutha Pandian, S., 2018. Fukugiside, a biflavonoid from Garcinia travancorica inhibits biofilm formation of Streptococcus pyogenes and its associated virulence factors. Journal of Medical Microbiology67(9), pp.1391-1401.

Ashwinkumar Subramenium, G., Viszwapriya, D., Iyer, P.M., Balamurugan, K. and Karutha Pandian, S., 2015. covR mediated antibiofilm activity of 3-furancarboxaldehyde increases the virulence of Group A Streptococcus. PloS one10(5), p.e0127210.

Figure 7. It is better to indicate the scale bar on microphotographs. The letter designations are incomprehensible, it is better to highlight these areas somehow.

Ans: The scale bars are included and the designations have been made clear.

Discussions. In the description of histostreses of the liver, kidneys and gills, you speak of the absence of obvious histopathological changes. So there were some histological changes compared to control samples?

Ans: No there was no histological changes at the tested concentrations as compared to control samples.

DOIs for articles are not indicated in the list of references.

Ans: DOIs of articles have been mentioned.

Reviewer 4 Report

This paper reports the antbiofilm and antivirulence activity of linalool, a major component of Hedychium larsenii.  The paper reports the identification and demonstration of relevant activity of linalool, along with clear characterisation of its biological role and mode of action as an antibiofilm agent, with low cytotoxicity.  Subject to some very minor corrections on the attached file, publication is recommended.

Author Response

Thanks for reviewing our papers. Modified according to your suggestions. 

All the changes notified in the pdf file has been made. Changes highlighted

Page 1 Line 20

Ans: We used the word impedes commonly for  micro-colony formation, mature biofilm architecture, surface coverage, and biofilm thickness.

Page 2 Line 54

Ans: hyphen has been added before polysaccharide

Page 2 Line 57

Ans: Affect removing has been changed to effect removal of.

Page 4 Line 199, 121

Ans: alamarBlueTM  is made uniform everywhere as  AlamarBlueTM

Page 4 Line 128

Ans: Alamarblue  has been changed to AlamarBlueTM

Representing is changed to showing.

Round 2

Reviewer 2 Report

The topic of this manuscript is interesting and important. The authors have revised the manuscript carefully.